# Adolescent individual, school, and neighborhood influences on young adult hypertension risk

Hoda S. Abdel Magid[1,2], Carly E. Milliren[3], Kathryn Rice[2], Nina Molanphy[2], Kennedy Ruiz[2], Holly C. Gooding[4,5], Tracy K. Richmond[6,7], Michelle C. Odden[1], Jason M. Nagata[8]*

1 Department of Epidemiology and Population Health, Stanford University, Stanford, California, United States of America, 2 Public Health Sciences Program, Santa Clara University, Santa Clara, California, United States of America, 3 Institutional Centers for Clinical and Translational Research, Boston Children's Hospital, Boston, Massachusetts, United States of America, 4 Division of General Pediatrics and Adolescent Medicine, Department of Pediatrics, Emory University School of Medicine, Atlanta, Georgia, United States of America, 5 Children's Healthcare of Atlanta, Atlanta, Georgia, United States of America, 6 Division of Adolescent and Young Adult Medicine, Boston Children's Hospital, Boston, Massachusetts, United States of America, 7 Department of Pediatrics, Harvard Medical School, Boston, Massachusetts, United States of America, 8 Division of Adolescent & Young Adult Medicine, University of California, San Francisco, San Francisco, California, United States of America

* jason.nagata@ucsf.edu

**Data Availability Statement:** This study analyses restricted-use data from Add Health. Persons interested in obtaining Data Files from Add Health should contact Add Health, The University of North

## Abstract

### Background

Geographic and contextual socioeconomic risk factors in adolescence may be more strongly associated with young adult hypertension than individual-level risk factors. This study examines the association between individual, neighborhood, and school-level influences during adolescence on young adult blood pressure.

### Methods

Data were analyzed from the National Longitudinal Study of Adolescent to Adult Health (1994–1995 aged 11–18 and 2007–2008 aged 24–32). We categorized hypertension as systolic blood pressure $\geq$140 mm Hg and/or diastolic blood pressure $\geq$90 mm Hg. Secondary outcomes included continuous systolic and diastolic blood pressure. We fit a series of cross-classified multilevel models to estimate the associations between young adulthood hypertension with individual-level, school-level, and neighborhood-level factors during adolescence (i.e., fixed effects) and variance attributable to each level (i.e., random effects). Models were fit using Bayesian estimation procedures. For linear models, intra-class correlations (ICC) are reported for random effects.

### Results

The final sample included 13,911 participants in 128 schools and 1,917 neighborhoods. Approximately 51% (7,111) young adults were hypertensive. Individual-level characteristics —particularly older ages, Non-Hispanic Black race, Asian race, male sex, BMI, and current

Carolina at Chapel Hill, Carolina Population Center, 206 W. Franklin Street, Chapel Hill, NC 27516-2524 (addhealth_contracts@unc.edu). Further information on how to obtain the Add Health data files is available on the Add Health website (http://www.cpc.unc.edu/addhealth). The authors did not receive special access privileges to the data that others would not have.

**Funding:** J.M.N. is supported by the National Heart, Lung, and Blood Institute (K08HL159350) and the American Heart Association (CDA34760281).

**Competing interests:** Potential Conflicts of Interest: Michelle Odden is a consultant for Cricket Health, Inc. The remaining authors have indicated no conflicts of interest to disclose. This does not alter our adherence to PLOS ONE policies on sharing data and materials.

smoking—were associated with increased hypertension. Non-Hispanic Black (OR = 1.21; 95% CI: 1.03–1.42) and Asian (OR = 1.28; 95% CI: 1.02–1.62) students had higher odds of hypertension compared to non-Hispanic White students. At the school level, hypertension was associated with the percentage of non-Hispanic White students (OR for 10% higher = 1.06; 95% CI: 1.01–1.09). Adjusting for individual, school, and neighborhood predictors attenuated the ICC for both the school (from 1.4 null to 0.9 fully-adjusted) and neighborhood (from 0.4 to 0.3).

## Conclusion

We find that adolescents' schools and individual-level factors influence young adult hypertension, more than neighborhoods. Unequal conditions in school environments for adolescents may increase the risk of hypertension later in life. Our findings merit further research to better understand the mechanisms through which adolescents' school environments contribute to adult hypertension and disparities in hypertension outcomes later in life.

## Introduction

High blood pressure and hypertension are growing problems in adolescents and young adults. The estimated prevalence of hypertension in the United States in 2017–2018 was 6–10% among adolescents (10–17 years) and 22.4% among young adults (18–39 years) [1–5]. The prevalence in adolescents is as high as 30% in adolescent boys with obesity and 23–30% in girls with obesity [6]. National statistics indicate that nearly 1 in 4 young adults in the U.S. experience elevated blood pressure [7, 8]. Hypertension is higher among young adult men than young adult women aged 18–39 (31.2% compared with 13.0%) [5, 9].

The study of hypertension risk factors in adolescence and young adulthood has many important public health implications. Hypertension, one of the major modifiable risk factors for cardiovascular disease (CVD), is established early in life. In a meta-analysis of 50 cohort studies, data from diverse populations show that blood pressure tracks from childhood into adulthood [10]. In addition, hypertension that begins in childhood is associated with adverse cardiac changes and vascular damage that in turn is associated with premature cardiovascular disease in adulthood [11]. Nevertheless, studies are limited examining whether adolescent risk factors related to sociocultural contexts (e.g., schools and neighborhoods) are associated with young adulthood hypertension; and thus, evidence is required to fill this gap and inform hypertension interventions in adolescence.

Studies have shown that schools and neighborhoods—the two contexts in which adolescents spend most of their time—have important bearings on cardiovascular risk factors including hypertension, diabetes, and obesity [12–17]. Cross-sectional and longitudinal evidence demonstrates that schools are more salient than neighborhoods in explaining variation in weight gain and body mass index (two important hypertension risk factors), and that schools provide direct opportunity and support for dietary intake and exercise. However, most studies face three key limitations: (1) they use single-level analysis, which cannot capture hypertension risk at a contextual level; (2) they examine one context at a time, making it difficult to compare the relative influences of multiple contexts; and (3) they examine the association cross-sectionally during adolescence and estimate the relationship between contextual risk factors and hypertension only in adolescence, and not young adulthood. To our knowledge, no prior

study has integrated these multilevel contexts during adolescence and compared their long-term influences on hypertension in young adulthood.

This study uses data from the National Longitudinal Study of Adolescent to Adult Health (Add Health) to investigate whether school and neighborhood contexts and their characteristics during adolescence are associated with the likelihood of hypertension in young adulthood. We hypothesized that higher socioeconomic status at both the school-level and neighborhood-level would predict lower hypertension risk in young adulthood. We hypothesized that school factors would be more strongly associated with young adult hypertension than neighborhood factors [7, 15].

## Methods

### Data collection

The National Longitudinal Study of Adolescent to Adult Health (Add Health) is a nationally representative cohort of adolescents in the U.S. who have been followed from adolescence through adulthood to identify social, behavioral, and biological determinants of health across the life course [18]. The Add Health study design is coordinated by the Carolina Population Center, as detailed elsewhere [19]. We obtained Institutional Review Board approval to conduct secondary analyses of the Add Health data using deidentified data obtained under an Add Health Restricted-Use Data Contract at the University of California, San Francisco.

### Participants

Add Health longitudinally follows a nationally representative sample of adolescents in grades 7 to 12 and ages 11–18 at baseline (Wave I; interviewed 1994–1995; N = 20,745) into adulthood. A sample of 80 high schools and 52 feeder middle schools from the United States was selected to ensure representation of U.S. schools with respect to region of country, urbanicity, school size, school type, and ethnicity. The current study uses in-home interview data from Wave I and Wave IV (aged 24–32 years; interviewed 2007–2008; N = 15,701). Of respondents who participated in both waves (N = 15,701), we excluded those missing measured blood pressure at Wave IV (n = 334); school contextual data (n = 910); neighborhood contextual data (n = 6); or individual SES measures (n = 540), resulting in a final analytic sample of 13,911 respondents.

### Outcomes

We constructed all outcome variables from the Wave IV in-home interview. After the interview, participants rested in a seated position for five minutes, after which three measures of blood pressure were recorded. Trained and certified Add Health field interviewers followed a computer-assisted data collection protocol to record participants' blood pressure. Interviewers measured blood pressure using an appropriately sized arm cuff and an automatic oscillometric monitor approved by the British Hypertension Society (BP 3MC1-PC_IB; MicroLife USA, Inc., Dunedin, FL) [20]. Three blood pressure measurements were taken at 30-second intervals from the right arm with the patient in the resting, seated position after 5 minutes of rest. The second and third measurements were double-entered and then averaged to give the final blood pressure recorded, which we used in the present study to measure hypertension status. After blood pressure measurement, the interviewer inventoried and recorded antihypertensive medications (beta-adrenergic blockers; calcium channel blockers; angiotensin converting enzyme inhibitors; angiotensin II receptor blockers; centrally or peripherally acting anti-adrenergics;

vasodilators; thiazide diuretics; antihypertensive combinations) used by participants within the preceding four weeks.

In primary analyses, we classified participants as hypertensive according to the Seventh Report of the Joint National Committee on Prevention, Detection, Evaluation and Treatment of High Blood Pressure—if they had an average measured systolic blood pressure (SBP) $\geq$ 140 mmHg, an average measured diastolic blood pressure $\geq$ 90 (DBP) mmHg, or use of antihypertensive medications [20]. In secondary analyses, we applied a more recent definition of hypertension based on the 2017 American College of Cardiology/American Heart Association Hypertension Guideline definition, and classified participants as hypertensive if they had an average SBP $\geq$ 130 mmHg and/or an average DBP $\geq$ 80 (DBP) mmHg, or use of antihypertensive medications [21]. Mean arterial pressure (MAP) was conventionally approximated as the weighted sum of systolic and diastolic blood pressure, using the following formula

$$MAP = \frac{1}{3}SBP + \frac{2}{3}DBP$$

where the weights for SBP (1/3) and DBP (2/3) reflect the typical contributions of ventricular systole and diastole to the duration of the cardiac cycle.

## Individual variables

We constructed individual covariates using data from the Wave I in-home interview, including adolescents' biological sex (male, female), race/ethnicity (non-Hispanic Black, Hispanic, Asian and Pacific Islander, Other, Multiracial, and non-Hispanic White). At the individual-level, SES was determined based on parental education and receipt of public assistance. We used data from either the youth or caregiver interview to capture receipt of public assistance (mother currently receiving public assistance, such as welfare or not) and highest level of parental education (defined as the maximum level of education by the resident mother, resident father, or resident step-father/partner (no high school diploma or equivalent; completed high school or equivalent; completed some college, trade school or a 2-year degree; completed equivalent 4-year college degree or above). Height and weight were measured by trained interviewers at Wave IV. Young adult body mass index (BMI) at Wave IV was calculated as the ratio of weight in kilograms over height in meters squared. Age at Wave IV (in years) was calculated from the date of Wave IV in-home interview and participant's date of birth.

## School variables

We constructed school-level covariates using data from the Wave I data. Using the survey of the full sample of schools, at the school-level, we created a continuous measure of school-level SES by aggregating individual-level data. Use of individual-level data was required as information about school-level SES was not directly available. We calculated the proportion of students within each school whose mother had received public assistance or had a college degree.

## Neighborhood variables

We constructed neighborhood-level covariates using data from the Wave I data. At the neighborhood level, we used data from the 1990 Census to create a neighborhood-level SES measure indicating the proportion of residents within each neighborhood who had received public assistance or had a college degree. We also calculated the proportion of students in either the school or the neighborhood who were White.

## Statistical analysis

We examined bivariate associations of individual-, school-, and neighborhood-level characteristics by Wave IV hypertension status. Differences in demographics and SES by hypertension were examined using two-sample *t*-tests for continuous variables and chi-square tests for categorical variables. School- and neighborhood-level demographics and SES are summarized using means (standard deviations). All tests were performed at an alpha-level of 0.05.

We constructed a series of cross-classified multilevel models (CCMM) to estimate the associations between Wave IV hypertension, systolic blood pressure, and mean arterial pressure with individual-level, school-level, and neighborhood-level factors (i.e., fixed effects) and the proportion of variance in Wave IV outcomes attributable to each level (i.e., random effects) [22]. The major advantage of CCMM compared with traditional multilevel models is that it allows for estimation of the effects of multiple non-nested contexts (e.g., students may attend schools outside of their neighborhoods and schools may draw students from multiple neighborhoods).

To parse out the effects of individual-level, neighborhood-level, and school-level contributions on hypertension, SBP, DBP, and MAP we ran logistic and linear regression models with model-building proceeding in a number of steps. We first examined the independent contributions of neighborhood and school contexts on the outcome using two-level hierarchical null (or unconditional) models. These models were fit by including individuals nested within either the school- or neighborhood-level. Next, school and neighborhood contexts were examined simultaneously by allowing for cross-classification of the two contexts. A total of four null models were fit including (1) individual-only, (2) individual and school, (3) individual and neighborhood, and (4) individual, school, and neighborhood. Subsequent models incorporated this cross-classification of school and neighborhood and the adjustment for other predictors via the following model equation. For example, SBP (denoted *y*) for an adolescent in the study (denoted *i*) nested in a given school (denoted *j*) and neighborhood (denoted *k*) was modeled as:

$$Y_{i(jk)} = \boldsymbol{\beta}_0 + \boldsymbol{\beta}_{xi} + \boldsymbol{\beta}_{xij} + \boldsymbol{\beta}_{xik} + \boldsymbol{u}_{0j} + \boldsymbol{u}_{0k} + \boldsymbol{e}_{0i(jk)}$$

with the following fixed effect parameters: $\beta_0$ refers to the overall mean SBP or MAP *y* across all schools and neighborhoods, $\beta_{xi}$ refers to the vector of individual-level covariates, $\beta_{xij}$ refers to the vector of school-level covariates, and $\beta_{xik}$ refers to the vector of neighborhood-level covariates. Random effect parameters included the following: $e_{0i(jk)}$ refers to the individual-level random effect variance parameter for the individual within the combination of school *j* and neighborhood *k*, $u_{0j}$ is the variance at the school-level and $u_{0k}$ is the variance at the neighborhood-level. A series of five adjusted cross-classified models were fitted. Model 1 adjusted for individual-level predictors including age, sex, race/ethnicity, parental education, and parental receipt of public assistance. Model 2 included individual-level predictors as well as the following school-level predictors: percentage of students of non-Hispanic White race, percentage of students whose parents receive public assistance and percentage of students whose parents have a college degree. Model 3 included individual predictors plus neighborhood-level predictors from the Census: percentage of residents' White race, percentage of residents receiving public assistance and percentage of residents with a college degree. Model 4 presents adjusted model all socioeconomic individual-, school, and neighborhood-level predictors. Model 5 presents the fully-adjusted model, which included all individual-, school-, and neighborhood-level predictors including individual BMI and smoking. All models for MAP, DBP and SBP additionally adjusted for self-reported use of antihypertensive medications.

For linear regression models predicting SBP, DBP, and MAP we report parameter estimates (β) and 95% credible intervals (CI) for fixed effects, parameter estimates (95% CI) for intercepts while variance estimates (95% CI) and intra-class correlations (ICC) are reported for random effects. ICCs allow for comparison of variance parameters across contextual levels and are interpreted as the percent of variance attributable to a given level. For logistic models predicting hypertension, we present odds ratios (OR) and 95% credible intervals for fixed effects, parameter estimates (95% CI) for intercepts, and variance estimates (95% CI) and ICC for random effects [23]. Model fit was evaluated using the deviance information criterion (DIC), which is a test statistic produced by the MCMC procedure that refers to the model complexity and "badness of fit" with higher DIC values indicate a poorer fitting model [24].

Models were fit using MLwiN (version 3.00; Birmingham, UK) via Stata's *runmlwin* command. MLwiN uses Bayesian estimation procedures using Markov Chain Monte Carlo (MCMC) methods with non-informative priors and a Metropolis-Hastings sampling algorithm allowing for simultaneous modeling of non-hierarchically nested contexts [24–27]. All univariate and bivariate analyses were preformed using Stata version 16 (College Station, TX).

## Results

The final analytic sample included a total of 13,911 participants from Wave IV from 128 schools and 1,917 neighborhoods. As shown in Table 1, the mean age of the analytic sample in Wave IV was 28.9 (SD = 1.7), 53.1% of the participants were non-Hispanic White, 20% were non-Hispanic Black, and 16% were Hispanic. The mean age of the analytic sample in Wave I was 15.6 (SD = 1.7). Participants' average SBP, DBP, and MAP were 124.5 (13.6), 79.0 (10.2), and 94.2 (10.7) mmHg, respectively; 509 (3.7%) of Wave IV study sample reported use of anti-hypertensive medications. Of the 13,911 Wave IV participants included in this study, 7,111 (51%) young adults were classified as hypertensive. For all outcomes, S3 Table presents results for null cross-classified multilevel models.

### Hypertension (140/90 mmHg)

In the null cross-classified model (Model 4, S3 Table), the random effects for school- and neighborhood-levels were 5% and 1% respectively. Table 2 shows the series of adjusted cross-classified models predicting hypertension (140/90 mmHg) among young adults in Wave IV. In the model adjusting only for individual factors (Model 1), the random effects for school- and neighborhood-levels were 1.08% and 0.3%, respectively. These indicate that the majority of variation in hypertension is due to individual or unmeasured variation, with a small percentage attributable to the school and a negligible percentage to the neighborhood. When school-level predictors were added to the model (Model 2), the school-level variance slightly increased to 1.2% attributable to school while the neighborhood variance remained stable with 0.3% of the variance being attributable to the neighborhood. Model 3 introduces neighborhood predictors into the individual-only model, and variance contributions of the school and neighborhood were similar to Model 2 (1.2% and 0.3%, respectively). In the fully-socioeconomic and fully-adjusted CCMM (Model 4) accounting for individual, school, and neighborhood-level predictors, ICCs for the school and neighborhood decreased to 0.9 and 0.3%. Comparing the variance parameters from the fully-adjusted model (Table 2, Model 5) to the null model (S3 Table, Model 4), adjusting for individual, school and neighborhood predictors attenuated the random effects for both the school (from 1.4 null to 0.9 fully-adjusted) and neighborhood (from 0.4 to 0.3).

In the fully-adjusted CCMM, we found significant associations between hypertension and individual-level fixed effects for age, female sex, race/ethnicity, BMI, and current smoking. For

**Table 1. Individual-, school-, and neighborhood-level Wave I (1994–1995) characteristics of participants in young adulthood at Wave IV (2008–2009; N = 13,926) of the National Longitudinal Study of Adolescent to Adult Health.**

| N (%) | Wave IV Total Sample (N = 13,926) | Wave IV Hypertension (N = 2,881) | Wave IV No Hypertension (N = 11,045) | |
|---|---|---|---|---|
| **Individual-level (N = 13,926)** | | | | **P-Value** |
| Age (years), Mean (SD) | 28.94 (1.72) | 29.14 (1.71) | 28.89 (1.72) | P<0.001 |
| Sex | | | | P<0.001 |
| Female | 7,373 (53) | 1,028 (14) | 6,345 (86) | |
| Male | 6,553 (47) | 1,853 (29) | 4,700 (71) | |
| Race/ethnicity | | | | P<0.001 |
| Non-Hispanic White | 7,395 (53) | 1,482 (51) | 5,913 (80) | |
| Non-Hispanic Black | 2,831 (20) | 669 (23) | 2,162 (76) | |
| Asian | 768 (6) | 164 (6) | 604 (79) | |
| Hispanic | 2,190 (16) | 407 (14) | 1,783 (81) | |
| Other | 172 (1) | 41 (1) | 131 (76) | |
| Multiracial | 570 (4) | 118 (4) | 452 (79) | |
| BMI, kg/m$^2$ (WIV) | | | | P<0.001 |
| Under Weight (<18.5) | 192 (1) | 16 (0.6) | 176 (1.6) | |
| Normal Weight (18.5–24.9) | 4,395 (32) | 459 (16) | 3,936 (35) | |
| Overweight (25–29.9) | 4.199 (30) | 818 (29) | 3,381 (30.6) | |
| Obese (≥30) | 5,125 (37) | 1,568 (55) | 3,546 (32) | |
| Unknown | 10 (0.1) | 4 (0.1) | 6 (0.05) | |
| Current Smoking (WIV) | | | | P<0.001 |
| No | 8,916 (64) | 1,735 (61) | 7,181 (65) | |
| Yes | 4,887 (35) | 1,101 (38) | 3,786 (34) | |
| Unknown | 108 (1) | 30 (1) | 78 (1) | |
| Antihypertensive medications | | | | |
| No | 13,417 (96) | 2,372 (82) | 11,045 (100) | |
| Yes | 509 (4) | 509 (18) | 0 (0) | |
| Parent Receipt of Public Assistance | | | | P = 0.18 |
| No | 12,703 (91) | 2,605 (90) | 10,098 (91) | |
| Yes | 1,223 (9) | 276 (10) | 947 (9) | |
| Parental Education | | | | P = 0.30 |
| Less than high school | 1,664 (12) | 338 (11) | 1,326 (12) | |
| High school graduate/GED | 3,611 (26) | 815 (28) | 2,796 (25) | |
| Some College | 4,154 (30) | 876 (30) | 3,278 (29) | |
| College graduate or beyond | 4,497 (32) | 852 (29) | 3,645 (33) | |
| **School-level (N = 128)** | | | | |
| | **Mean (SD)** | **Median** | **Minimum** | **Maximum** |
| Percent of students Non-Hispanic White | 47.5 (25.5) | 55.0 | 0 | 85.9 |
| Percent of parents receiving public assistance | 10.4 (9.4) | 7.2 | 0 | 45.4 |
| Percent of parents with college degree | 31.7 (16.9) | 28.3 | 5.5 | 91.2 |
| **Neighborhood-level (N = 1,917)** | | | | |
| Percent of residents Non-Hispanic White | 67.1 (32.5) | 79.7 | 0 | 100 |
| Percent of residents receiving public assistance | 10.5 (9.6) | 7.2 | 0 | 61.8 |
| Percent of residents with college degree | 23.8 (14.6) | 20.3 | 1.1 | 77.8 |

**Table 2. Logistic cross-classified multilevel models (CCMM) predicting hypertension from individual-, school- and neighborhood-level factors in the National Longitudinal Study of Adolescent to Adult Health, Wave IV (WIV), 2008–2009 (N = 13,926).**

| Hypertension (140/90) | Model 1 | Model 2 | Model 3 | Model 4 | Model 5 |
|---|---|---|---|---|---|
| | Individual Cross-Classified | Individual and School Cross-Classified | Individual and Neighborhood Cross-Classified | Individual, School, and Neighborhood Cross-Classified | Individual, School, and Neighborhood Cross-Classified |
| **Fixed effect estimates Odds Ratios (95% Credible Interval)** | | | | | |
| Intercept | -3.55 (-3.94, -3.11) | -3.34 (-3.90, -2.98) | -3.38 (-4.15, -2.50) | -2.79 (-3.37, -2.11) | 0.04 (0.02, 0.10) |
| **Individual-level** | | | | | |
| Age, years (WIV) | 1.09 (1.07, 1.11) | 1.08 (1.06, 1.10) | 1.09 (1.06, 1.12) | 1.06 (1.04, 1.09) | 1.05 (1.02, 1.07) |
| Female | 0.41 (0.37, 0.44) | 0.41 (0.37, 0.44) | 0.41 (0.37, 0.44) | 0.41 (0.37, 0.44) | 0.40 (0.36, 0.44) |
| Race/ethnicity | | | | | |
| Non-Hispanic White | REF | REF | REF | REF | REF |
| Non-Hispanic Black | 1.28 (1.13, 1.42) | 1.35 (1.17, 1.55) | 1.19 (1.02, 1.37) | 1.21 (1.03, 1.42) | 1.21 (1.03, 1.42) |
| Asian | 1.06 (0.86, 1.30) | 1.14 (0.91, 1.40) | 1.03 (0.83, 1.24) | 1.11 (0.88, 1.39) | 1.28 (1.02, 1.62) |
| Hispanic | 0.93 (0.80, 1.07) | 0.97 (0.83, 1.14) | 0.90 (0.77, 1.04) | 0.96 (0.82, 1.13) | 0.92 (0.77, 1.08) |
| Other | 1.23 (0.82, 1.78) | 1.28 (0.84, 1.86) | 1.21 (0.81, 1.75) | 1.15 (0.84, 1.73) | 1.17 (0.79, 1.66) |
| Multiracial | 1.14 (0.99, 1.32) | 1.17 (0.92, 1.26) | 1.10 (0.88, 1.35) | 1.14 (0.91, 1.40) | 1.13 (0.90, 1.40) |
| Parent receipt of public assistance | 1.11 (0.96, 1.30) | 1.09 (0.92, 1.26) | 1.08 (0.91, 1.26) | 1.07 (.92, 1.23) | 1.06 (0.89, 1.24) |
| Parental Education | | | | | |
| Less than high school | REF | REF | REF | REF | REF |
| High school graduate / GED | 1.14 (0.98, 1.32) | 1.13 (0.98, 1.29) | 1.14 (0.98, 1.32) | 1.12 (0.96, 1.28) | 1.11 (0.94, 1.30) |
| Some college | 1.03 (0.88, 1.18) | 1.02 (0.89, 1.17) | 1.03 (0.87, 1.19) | 1.01 (0.87, 1.17) | 1.02 (0.85, 1.21) |
| College graduate or beyond | 0.92 (0.78, 1.07) | 1.00 (1.00, 1.02) | 0.93 (0.78, 1.08) | 0.92 (0.78, 1.07) | 0.98 (0.81, 1.21) |
| BMI, kg/m$^2$ (WIV) | | | | | |
| Under or Normal Weight | - | - | - | - | REF |
| Overweight | - | - | - | - | 1.92 (1.70, 2.16) |
| Obese | - | - | - | - | 3.87 (3.47, 4.31) |
| Unknown | - | - | - | - | 9.64 (1.77, 28.01) |
| Current smoking (WIV) | | | | | |
| No | - | - | - | - | REF |
| Yes | - | - | - | - | 1.15 (1.04, 1.25) |
| Unknown | - | - | - | - | 1.36 (0.84, 2.04) |
| **School-level, per 10%** | | | | | |
| Percent of students Non-Hispanic White | - | 1.03 (1.00, 1.07) | - | 1.05 (1.02, 1.08) | 1.06 (1.01, 1.09) |
| Percent of parents receiving public assistance | - | 1.06 (0.96, 1.16) | - | 1.02 (0.93, 1.13) | 1.02 (0.92, 1.13) |
| Percent of parents with college degree | - | 0.98 (0.94, 1.04) | - | 0.97 (0.93, 1.02) | 0.98 (0.95, 1.03) |
| **Neighborhood-level, per 10%** | | | | | |
| Percent of residents Non-Hispanic White | - | - | 0.99 (0.97, 1.02) | 0.96 (0.93, 0.98) | 0.97 (0.94, 1.01) |
| Percent of residents receiving public assistance | - | - | 1.07 (0.96, 1.17) | 1.05 (0.96, 1.14) | 1.05 (0.95, 1.16) |
| Percent of residents with college degree | - | - | 0.99 (0.95, 1.04) | 1.01 (0.95, 1.06) | 1.05 (0.99, 1.10) |
| **Random effect and variance estimates (95% Credible Interval) [ICC, %]** | | | | | |

*(Continued)*

**Table 2.** (Continued)

| Hypertension (140/90) | Model 1 | Model 2 | Model 3 | Model 4 | Model 5 |
|---|---|---|---|---|---|
| | Individual Cross-Classified | Individual and School Cross-Classified | Individual and Neighborhood Cross-Classified | Individual, School, and Neighborhood Cross-Classified | Individual, School, and Neighborhood Cross-Classified |
| School | 0.04 (0.02–0.06) [1.08] | 0.04 (0.01–0.07) [1.20] | 0.04 (0.01–0.07) [1.20] | 0.03 (0.00–0.06) [0.90] | 0.03 (0.00, 0.06) [0.90] |
| Neighborhood | 0.01 (0.00–0.02) [0.30] | 0.01 (0.00–0.02) [0.30] | 0.01 (0.00–0.03) [0.30] | 0.01 (0.01–0.03) [0.30] | 0.01 (0, 0.03) [0.30] |
| Fit statistics (DIC) | 13669.94 | 13670.47 | 13671.14 | 13671.22 | 13033.32 |

every additional year in age, young adults had 1.05 higher odds of hypertension (95% CI: 1.02, 1.07). Females were less likely to have hypertension than males (OR = 0.40, 95% CI: 0.36, 0.44). Both Non-Hispanic Black (OR = 1.21; 95% CI: 1.03–1.42) and Asian (OR = 1.28; 95% CI: 1.02–1.62) students had higher odds of hypertension as compared to non-Hispanic White students. Compared to students with under or normal weight BMIs, students with overweight BMI (OR = 1.92; 95% CI: 1.70–2.16) or obese BMI (OR = 3.87; 95% CI: 3.47–4.31) had increased odds of hypertension. Moreover, current smokers had higher odds of hypertension (OR = 1.15; 95% CI: 1.04–1.25).

At the school level, we detected an association between hypertension and the percentage of non-Hispanic White students in the school (OR for 10% higher = 1.06; 95% CI: 1.01–1.09). Moreover, compared to the individual, school, and neighborhood only cross-classified models, the fully adjusted cross-classified models accounting for individual, school, and neighborhood fixed effects had the lowest DIC value indicating a stronger fitting model. There was no association with the percentage of students in the school whose parent had received public assistance or percentage of students in the school whose parents have a college degree and young adult hypertension. Neighborhood-level fixed effects were not associated with hypertension.

Results from cross-classified logistic models predicting a more recent definition of hypertension (SBP/DBP of ≥130/80 or use of antihypertensives) during young adulthood (Wave IV) are presented in S1 Table and were comparable to the findings from logistic regression with our primary hypertension classification. For example, we found significant associations between hypertension and individual-level fixed effects for age (OR = 1.06; 95% CI: 1.05–1.08), female sex (OR = 0.33; 95% CI:1.05–1.08), overweight BMI (OR = 1.87; 95% CI: 1.71–2.04) or obese BMI (OR = 3.65; 95% CI: 3.33–3.98), current smoking (OR = 1.11; 95% CI: 1.02–1.20). School and neighborhood-level fixed effects were not associated with hypertension (130/80 mmHg). Comparing the variance parameters from the fully-adjusted model (Table 3, Model 5) to the null model (S3 Table, Model 4), adjusting for individual, school and neighborhood predictors attenuated the variance contributions for both the school (from 1.4 null to 0.6 fully-adjusted) and neighborhood (from 0.4 to 0.3).

### Systolic and diastolic blood pressure blood pressure

In our null models predicting systolic blood pressure, not accounting for fixed effects at any level, individual level random effects accounted for 98.5% of the variance, school for 1.1% and neighborhoods for 0.4% (S3 Table, Model 4). With the inclusion of individual level fixed effects, the individual level random effects did not substantially increase (from 98.5% in the null model to 99% in the individual-only model). The same trend held for models adding school-level effects (school-level random effect: 0.7%) and adding neighborhood-level fixed

**Table 3. Linear cross-classified multilevel models (CCMM) predicting systolic blood pressure from individual-, school- and neighborhood-level factors in the National Longitudinal Study of Adolescent to Adult Health, Wave IV (WIV), 2008–2009 (N = 13,926).**

| Systolic Blood Pressure (mmHg) | Model 1 | Model 2 | Model 3 | Model 4 | Model 5 |
|---|---|---|---|---|---|
| | Individual Cross-Classified | Individual and School Cross-Classified | Individual and Neighborhood Cross-Classified | Individual, School, and Neighborhood Cross-Classified | Individual, School, and Neighborhood Cross-Classified |
| **Fixed effect estimates β (95% CI)** | | | | | |
| Intercept (SE) | 123.39 (119.34, 127.32) | 123.83 (119.53, 127.99) | 124.59 (120.32, 128.79) | 125.22 (120.84, 129.62) | 120.49 (116.35, 124.67) |
| **Individual-level** | | | | | |
| Age, years (WIV) | 0.21 (0.07, 0.34) | 0.20 (0.06, 0.33) | 0.21 (0.07, 0.34) | 0.18 (0.05, 0.31) | 0.11 (-0.02, 0.24) |
| Female | -9.93 (-10.36, -9.51) | -9.93 (-10.35, -9.51) | -9.94 (-10.35, -9.52) | -9.95 (-10.37, -9.53) | -9.50 (-9.89, -9.10) |
| Race/ethnicity | | | | | |
| Non-Hispanic White | REF | REF | REF | REF | REF |
| Non-Hispanic Black | 1.88 (1.25, 2.50) | 2.15 (1.45, 2.87) | 1.54 (0.76, 2.31) | 1.68 (0.91, 2.45) | 1.35 (0.59, 2.08) |
| Asian | 1.88 (-1.45, 0.63) | -0.16 (-1.28, 0.93) | -0.53 (-1.59, 0.52) | -0.25 (-1.35, 0.86) | 0.59 (-0.50, 1.65) |
| Hispanic | -0.69 (-1.41, 0.06) | -0.48 (-1.28, 0.33) | -0.84 (-1.62. -0.06) | -0.52 (-1.32, 0.27) | -0.96 (-1.68, -0.22) |
| Other | 1.24 (-0.62, 3.12) | 1.45 (-0.54, 3.41) | 1.14 (-0.84, 3.11) | 1.38 (-0.60, 3.27) | 0.98 (-0.90, 2.85) |
| Multiracial | 0.009 (-1.07, 1.09) | 0.15 (-0.93, 1.27) | -0.08 (-1.15, 1.02) | 0.07 (-1.03, 1.18) | -0.29 (-1.34, 0.74) |
| Parental education | | | | | |
| Less than high school | REF | REF | REF | REF | REF |
| High school graduate / GED | 0.026 (-0.37, 1.17) | 0.35 (-0.42, 1.13) | 0.17 (-0.61, 0.97) | 0.39 (-0.35, 1.17) | 0.31 (-0.42, 1.03) |
| Some college | 0.39 (-0.95, 0.57) | -0.19 (0.99, 0.58) | 0.45 (-0.32, 1.24) | -0.13 (-0.91, 0.65) | -0.05 (-0.79, 0.69) |
| College graduate or beyond | -0.19 (-1.86, 0.27) | -1.01 (-1.78, -0.22) | -0.05 (-0.85, 0.72) | -0.86 (-1.67, -0.06) | -0.25 (-1.01, 0.48) |
| Parent receipt of public assistance | 0.26 (-0.52, 1.05) | 0.28 (-0.51, 1.07) | 0.02 (-0.61, 0.97) | 0.21 (-0.59, 1.00) | 0.14 (-0.61, 0.92) |
| Antihypertensive medication | 7.58 (6.45, 8.69) | 7.57 (6.45, 8.70) | 7.55 (6.42, 8.67) | 7.52 (6.38, 8.65) | 5.63 (4.55, 5.55) |
| BMI, kg/m$^2$ (WIV) | | | | | |
| Under or Normal Weight | | | | | REF |
| Overweight | | | | | 5.03 (4.52, 5.55) |
| Obese | | | | | 9.42 (8.93, 9.89) |
| Unknown | | | | | 16.96 (9.66, 24.11) |
| Current smoking (WIV) | | | | | |
| No | | | | | REF |
| Yes | | | | | 1.07 (0.64, 1.49) |
| Unknown | | | | | 0.95 (-1.33, 3.22) |
| **School-level, per 10%** | | | | | |
| Percent of students Non-Hispanic White | | 0.09 (-0.06, 0.25) | | 0.17 (-0.01, 0.36) | 0.15 (-0.01, 0.15) |
| Percent of parents receiving public assistance | | -0.19 (-0.70, 0.32) | | -0.19 (-0.74, 0.35) | -0.29 (-0.80, -0.29) |
| Percent of parents with college degree | | -0.17 (-0.40, 0.07) | | -0.08 (-0.35, 0.18) | 0.02 (-0.22, 0.17) |
| **Neighborhood-level, per 10%** | | | | | |
| Percent of students Non-Hispanic White | | | -0.06 (-0.19, 0.06) | -0.14 (-0.29, 0.01) | -0.07 (-0.21, -0.07) |
| Percent of parents receiving public assistance | | | -0.03 (-0.50, 0.44) | 0.06 (-0.43, 0.56) | 0.16 (-0.29, 0.16) |

(*Continued*)

**Table 3.** (Continued)

| Systolic Blood Pressure (mmHg) | Model 1 | Model 2 | Model 3 | Model 4 | Model 5 |
|---|---|---|---|---|---|
| | **Individual Cross-Classified** | **Individual and School Cross-Classified** | **Individual and Neighborhood Cross-Classified** | **Individual, School, and Neighborhood Cross-Classified** | **Individual, School, and Neighborhood Cross-Classified** |
| Percent of parents with college degree | | | -0.30 (-0.54, -0.04) | -0.23 (-0.50, 0.03) | -0.02 (-0.26, -0.02) |
| **Random effect and variance estimates (95% Credible Interval) [ICC, %]** | | | | | |
| Individual | 157 (153, 160) [99] | 156 (152, 160) [99] | 156 (152, 160) [99] | 156 (152, 160) [99] | 141 (138, 145) [99] |
| School | 1 (0.6, 2.2) [0.9] | 1 (0, 2) [0.7] | 1 (0, 1.9) [0.6] | 1.16 (0.5, 2.1) [0.7] | 0.9 (0.3, 1.6) [0.6] |
| Neighborhood | 0.1 (0, 0.4) [0.1] | 0.4 (0, 1.9) [0.3] | 0.4 (0, 1.7) [0.4] | 0.06 (0.01, 0.2) [0.3] | 0.1 (0, 0.4) [0.1] |
| Fit statistics (DIC) | 109841.9 | 109850.1 | 109847 | 109839.5 | 108468.56 |

effects (neighborhood-level random effect: 0.4%). In null models predicting diastolic blood pressure, not accounting for fixed effects at any level, individual level random effects accounted for 99% of the variance, school for 0.1% and neighborhoods for 0.1% (S3 Table, Model 4). Similar to results for systolic blood pressure, with the inclusion of fixed effects at each level, the individual, school, or neighborhood-level random effects did not substantially change (Table 4). We then examined the associations of characteristics of individuals, schools and neighborhoods with young adult systolic blood pressure, and found that compared to hypertension, many significant relationships did not remain. For example, students with over-weight BMI (β = 5.03; 95% CI: 4.52–5.55) or obese BMI (β = 9.42; 95% CI: 8.93–9.89) had higher systolic blood pressure. At the neighborhood level, in the fully adjusted individual, school, and neighborhood cross-classified model, (Table 3, Model 5) percent of students Non-Hispanic White was associated with lower systolic blood pressure (β = -0.07; 95% CI: -0.21, -0.07). Other neighborhood-level factors and school-level fixed effects were not associated with either systolic or diastolic blood pressure.

Results from cross-classified linear models predicting MAP during young adulthood (Wave IV) are presented in S2 Table, and were comparable to the findings from linear regressions of systolic and diastolic blood pressure (Tables 3 and 4).

## Discussion

To our knowledge, this is the first study to compare the influence of individual adolescent factors, schools and neighborhoods on young adult blood pressure and hypertension outcomes simultaneously. This study adds to previous literature on contextual influences on adolescent and young adult development by exploring the relative contributions of both school-level and neighborhood-level socioeconomic characteristics to young adult blood pressure and hypertension using a school-based sample of US adolescents. We found that the variation in hypertension was largely explained at the individual level with only small but significant contributions at the school- and neighborhood-level. These results suggest that the between-level variation in hypertension was due largely to the observed individual characteristics across schools and neighborhoods, and that more of the variability in hypertension was attributable to the school-level characteristics than the neighborhood-level characteristics.

Individual-level characteristics, particularly older ages, Non-Hispanic Black race, Asian race, male sex, BMI, and current smoking, were associated with increased hypertension

**Table 4. Linear cross-classified multilevel models (CCMM) predicting diastolic blood pressure from individual-, school- and neighborhood-level factors in the National Longitudinal Study of Adolescent to Adult Health, Wave IV (WIV), 2008–2009 (N = 13,926).**

| Diastolic Blood Pressure (mmHg) | Model 1 | Model 2 | Model 3 | Model 4 | Model 5 |
|---|---|---|---|---|---|
| | Individual Cross-Classified | Individual and School Cross-Classified | Individual and Neighborhood Cross-Classified | Individual, School, and Neighborhood Cross-Classified | Individual, School, and Neighborhood Cross-Classified |
| **Fixed effect estimates β (95% CI)** | | | | | |
| Intercept (SE) | 69.61 (66.45, 72.73) | 69.98 (66.56, 73.27) | 70.27 (66.91, 73.60) | 70.79 (67.29, 74.21) | 67.74 (64.41, 71.15) |
| **Individual-level** | | | | | |
| Age, years (WIV) | 0.40 (0.29, 0.51) | 0.39 (0.29, 0.50) | 0.40 (0.30, 0.51) | 0.38 (0.28, 0.49) | 0.35 (0.24, 0.45) |
| Female | -4.75 (-5.08, -4.42) | -4.74 (-5.07, -4.42) | -4.75 (-5.07, -4.42) | -4.75 (-5.08, -4.43) | -4.48 (-4.79, -4.16) |
| Race/ethnicity | | | | | |
| White | REF | REF | REF | REF | REF |
| Black | 1.02 (0.53, 1.51) | 1.20 (0.65, 1.76) | 0.88 (0.28, 1.49) | 0.95 (0.35, 1.55) | 0.76 (0.17, 1.35) |
| Asian | 0.70 (-0.10, 1.52) | 0.89 (0.03, 1.74) | 0.67 (-0.16, 1.48) | 0.85 (0.01, 1.72) | 1.41 (0.55, 2.26) |
| Hispanic | -0.34 (-0.91, 0.24) | -0.20 (-0.81, 0.44) | -0.40 (-0.99, 0.19) | -0.24 (-0.86, 0.37) | -0.51 (-1.08, 0.07) |
| Other | 0.31 (-1.14, 1.77) | 0.42 (-1.11, 1.94) | 0.28 (-1.26, 1.81) | 0.39 (-1.14, 1.86) | 0.13 (-1.35, 1.61) |
| Multiracial | 0.08 (-0.75, 0.93) | 0.19 (-0.65, 1.06) | 0.06 (-0.76, 0.92) | 0.16 (-0.71, 1.02) | -0.07 (-0.91, 0.74) |
| Parental education | | | | | |
| Less than high school | REF | REF | REF | REF | REF |
| High school graduate / GED | 0.28 (-0.31, 0.88) | 0.26 (-0.33, 0.87) | 0.32 (-0.28, 0.92) | 0.29 (-0.29, 0.88) | 0.21 (-0.36, 0.78) |
| Some college | 0.01 (-0.57, 0.61) | 0.05 (-0.57, 0.65) | 0.10 (-0.52, 0.71) | 0.08 (-0.52, 0.69) | 0.12 (-0.46, 0.70) |
| College graduate or beyond | -0.66 (-1.28, -0.04) | -0.55 (-1.14, 0.07) | -0.48 (-1.09, 0.15) | -0.46 (-1.09, 0.16) | -0.07 (-0.67, 0.51) |
| Parent receipt of public assistance | 0.36 (-0.25, 0.98) | 0.32 (-0.29, 0.93) | 0.31 (-0.31, 0.92) | 0.28 (-0.33, 0.90) | 0.22 (-0.37, 0.83) |
| Antihypertensive medication | 5.29 (4.42, 6.16) | 5.28 (4.41, 6.16) | 5.27 (4.40, 6.14) | 5.25 (4.27, 6.13) | 4.05 (3.21, 4.90) |
| BMI, kg/m$^2$ (WIV) | | | | | |
| Under or Normal Weight | | | | | REF |
| Overweight | | | | | 2.80 (2.39, 3.21) |
| Obese | | | | | 5.99 (5.61, 6.36) |
| Unknown | | | | | 8.04 (2.27, 13.69) |
| Current smoking (WIV) | | | | | |
| No | | | | | REF |
| Yes | | | | | 0.94 (0.60, 1.28) |
| Unknown | | | | | 0.06 (-1.74, 1.85) |
| **School-level, per 10%** | | | | | |
| Percent of students Non-Hispanic White | | 0.01 (-0.04, 0.21) | | 0.13 (-0.02, 0.28) | 0.11 (-0.02, 0.25) |
| Percent of parents receiving public assistance | | -0.05 (-0.46, 0.38) | | -0.04 (-0.48, 0.39) | -0.11 (-0.53, 0.31) |
| Percent of parents with college degree | | -0.22 (-0.41, -0.02) | | -0.16 (-0.38, 0.04) | -0.09 (-0.29, 0.11) |
| **Neighborhood-level, per 10%** | | | | | |
| Percent of residents Non-Hispanic White | | | -0.02 (-0.12, 0.09) | -0.07 (-0.19, 0.05) | -0.03 (-0.14, 0.08) |
| Percent of residents receiving public assistance | | | -0.01 (-0.36, 0.37) | 0.01 (-0.38, 0.39) | 0.06 (-0.30, 0.41) |
| Percent of residents with college degree | | | -0.24 (-0.43, -0.03) | -0.14 (-0.35, 0.08) | 0.01 (-0.20, 0.19) |

*(Continued)*

**Table 4.** (Continued)

| Diastolic Blood Pressure (mmHg) | Model 1 | Model 2 | Model 3 | Model 4 | Model 5 |
|---|---|---|---|---|---|
| | Individual Cross-Classified | Individual and School Cross-Classified | Individual and Neighborhood Cross-Classified | Individual, School, and Neighborhood Cross-Classified | Individual, School, and Neighborhood Cross-Classified |
| **Random effect and variance estimates (95% Credible Interval) [ICC, %]** | | | | | |
| Individual | 101.31 (92.3, 96.6) [0.9] | 94.48 (92.1, 96.6) [0.9] | 94.36 (92.1, 96.6) [0.9] | 94.46 (92.3, 96.7) [0.9] | 88.46 (86.42, 90.58) [0.9] |
| School | 1 (0.5, 1.5) [0.01] | 1 (0.5, 1.6) [0.01] | 1 (0.6, 1.6) [0.01] | 0.9 (0.4, 1.4) [0.02] | 0.70 (0.34, 1.16) [0.1] |
| Neighborhood | 0.04 (0, 0.3) [0.01] | 1 (0.55, 1.6) [0.01] | 1 (0.6, 1.6) [0.01] | 0.8 (0.5, 1.4) [0.01] | 0.07 (0.01, 0.31) [0.1] |
| Fit statistics (DIC) | 102829.1 | 102826.9 | 102829.1 | 102829.69 | 101919.40 |

among young adults. These findings at the individual-level add to the literature demonstrating that individual-level risk factors in adolescents influence hypertension risk later in life. Consistent with previous evidence, we found that hypertension risk increases with age and is higher for young adult men than women, and Black compared to White young adults [5, 9]. Moreover, this is consistent with the substantive body of literature indicating that Non-Hispanic Black Americans develop hypertension earlier in life than White Americans and provides further evidence that the racial/ethnic disparities in hypertension risk factors can appear as early as adolescence [28]. There is significant evidence showing racial/ethnic disparities in hypertension among young adults are linked to disparities in obesity, physical activity, and healthcare access, among other risk factors for hypertension. Moreover, these findings may be explained by adolescents' exposures to everyday discrimination and racism. Several studies have found associations between reports of discrimination and self-reported everyday discrimination with hypertension including a systematic review evaluating the association between perceived racial discrimination with hypertensive status and systolic, diastolic, and ambulatory blood pressure [29–31].

Similar to other studies of contextual influences on adolescents' cardiovascular risk factors, we found that school-level influences are related to adult health outcomes [13, 15, 32–34]. Of note, at the school level, we found that having a higher proportion of non-Hispanic White students was associated with higher hypertension risk into young adulthood 14 years later at follow-up. This finding is in the opposite direction for the findings for individual-level race/ethnicity with hypertension and neighborhood-level race/ethnicity for systolic blood pressure, which suggests an increased risk for students of color. This finding also suggests that unequal conditions for adolescents at the school level may increase the risk of hypertension later in life. This finding aligns with previous research showing the intersection between the social determinants of health and disparities by race/ethnicity are rooted in structural racism that results in inequitable access to resources required for health and well-being including uneven access to quality schools, better neighborhoods, and quality medical care [35]. Moreover, exclusionary policies such as redlining have had the effect of reducing the quality of local schools. This school-level finding may also reflect influences of other attributes of adolescents' school environment including the food environment and access to physical activity during school hours. For example, in a study using Add Health data, investigators found differences in physical activity levels in Hispanic and Non-Hispanic Black adolescents as compared to Non-Hispanic White adolescents and that these differences were largely attributable to the schools the

adolescents attended [36]. Relatedly, in an adjusted analysis of Study of Cardiovascular Risks in Adolescents (ERICA) of students enrolled in public and private schools located in urban and rural areas of Brazil, investigators found that consumption of meals prepared on the school premises was associated with adolescents' hypertension risk (OR = 0.79, 95% CI: 0.69–0.92), implying that the school food environment in adolescence may influence their cardiovascular health [37].

## Limitations

This study has limitations that merit acknowledgement. First, analyses are based on a study that selected adolescents using school-based sampling resulting in a large proportion of small neighborhoods. Although 45% of neighborhoods at Wave I contained a single respondent, prior work using Add Health has indicated no issue with bias in the random effect estimates as a result of small neighborhood sizes [38]. Second, limited school and neighborhood-level measures during adolescence were available and thus this study may miss other contextual attributes at the school and neighborhood level that may influence young adult hypertension risk measures of the built environment and access to green space. Nevertheless, Add Health is one of the few large, national samples of adolescents in the US that collected school and neighborhood-level data along with follow-up into young adulthood. Data were unweighted in these analyses because complex sample weighting techniques for CCMMs are not well-established. Nonetheless, strengths of the study included a large, national sample, and longitudinal study design. Given the discordance in young adult hypertension between NHANES and Add Health studies, some have questioned the accuracy and reliability of blood pressure in Add Health. However, one study found that, compared to NHANES, Add Health's terminal digit preference of blood pressure is infrequent, bias is low, short-term reliability is good to excellent, and comparable to that found in well-known, exam center-based studies of cardiovascular disease [2]. Therefore, our study's findings provide further evidence that the prevalence of hypertension among Add-Health Wave-IV participants indicates an unexpectedly high risk of cardiovascular disease among U.S. young adults and deserves further scrutiny [2].

## Conclusion

In conclusion, we find that adolescents' schools and individual-level factors influence young adult hypertension, more than neighborhoods. Our study contributes to the sparse literature examining multiple contextual contributors to young adult hypertension and indicate that the individual and school-level adolescent contexts may be the most important environments. Understanding the relative importance of these various contexts is important for developing targeted interventions to reduce hypertension risk factors in adolescents, hypertension in young adulthood, and cardiovascular disease later in life. Understanding these contexts can inform implementation strategies for hypertension prevention and health promotion efforts at the individual and school levels. Our findings merit further research to better understand the mechanisms through which adolescents' school environments contribute to adult hypertension and disparities in hypertension outcomes later in life.

## Supporting information

**S1 Table. Series of adjusted cross-classified multilevel models predicting hypertension (130/80) based on the 2017 ACC/AHA guidelines.**
(DOCX)

**S2 Table. Series of adjusted cross-classified multilevel models predicting mean arterial pressure (MAP), defined as the weighted sum of systolic and diastolic blood pressure.**
(DOCX)

**S3 Table. Null cross-classified multilevel models for all outcomes including hypertension (140/90), hypertension (130/80), systolic blood pressure, diastolic blood pressure, and mean arterial pressure (MAP).**
(DOCX)

## Acknowledgments

Add Health is directed by Robert A. Hummer and funded by the National Institute on Aging cooperative agreements U01 AG071448 (Hummer) and U01AG071450 (Aiello and Hummer) at the University of North Carolina at Chapel Hill. Waves I-V data are from the Add Health Program Project, grant P01 HD31921 (Harris) from Eunice Kennedy Shriver National Institute of Child Health and Human Development (NICHD), with cooperative funding from 23 other federal agencies and foundations. Add Health was designed by J. Richard Udry, Peter S. Bearman, and Kathleen Mullan Harris at the University of North Carolina at Chapel Hill.

## Author Contributions

**Conceptualization:** Hoda S. Abdel Magid, Carly E. Milliren, Jason M. Nagata.

**Data curation:** Hoda S. Abdel Magid, Carly E. Milliren, Jason M. Nagata.

**Formal analysis:** Hoda S. Abdel Magid, Carly E. Milliren.

**Funding acquisition:** Jason M. Nagata.

**Investigation:** Hoda S. Abdel Magid.

**Methodology:** Hoda S. Abdel Magid, Carly E. Milliren, Jason M. Nagata.

**Project administration:** Hoda S. Abdel Magid.

**Software:** Hoda S. Abdel Magid.

**Supervision:** Hoda S. Abdel Magid, Holly C. Gooding, Tracy K. Richmond, Michelle C. Odden, Jason M. Nagata.

**Visualization:** Hoda S. Abdel Magid.

**Writing – original draft:** Hoda S. Abdel Magid, Kathryn Rice, Nina Molanphy, Kennedy Ruiz, Jason M. Nagata.

**Writing – review & editing:** Hoda S. Abdel Magid, Carly E. Milliren, Holly C. Gooding, Tracy K. Richmond, Michelle C. Odden, Jason M. Nagata.

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
