## [Decision Letter · Decision Letter 0]

31 Jan 2022

PONE-D-21-33507Adolescent Individual, School, and Neighborhood Influences on Young Adult Hypertension RiskPLOS ONE

Dear Dr. Nagata,

Thank you for submitting your manuscript to PLOS ONE. After careful consideration, we feel that it has merit but does not fully meet PLOS ONE’s publication criteria as it currently stands. Therefore, we invite you to submit a revised version of the manuscript that addresses the points raised during the review process.

We look forward to receiving your revised manuscript.

Kind regards,

Giacomo Pucci

Academic Editor

PLOS ONE

Journal Requirements:

"I have read the journal's policy and the authors of this manuscript have the following competing interests: Michelle Odden is a consultant for Cricket Health, Inc. The remaining authors have indicated no conflicts of interest to disclose."

We note that you received funding from a commercial source: Cricket Health, Inc

Please include your amended Competing Interests Statement within your cover letter. We will change the online submission form on your behalf

Reviewers' comments:

Reviewer's Responses to Questions

**Comments to the Author**

1. Is the manuscript technically sound, and do the data support the conclusions?

Reviewer #1: Yes

Reviewer #2: Yes

2. Has the statistical analysis been performed appropriately and rigorously? 

Reviewer #1: Yes

Reviewer #2: Yes

3. Have the authors made all data underlying the findings in their manuscript fully available?

Reviewer #1: No

Reviewer #2: No

4. Is the manuscript presented in an intelligible fashion and written in standard English?

Reviewer #1: Yes

Reviewer #2: Yes

5. Review Comments to the Author

Reviewer #1: The manuscript considers secondary data analysis and fitting of cross-classified multilevel models (CCMM) to AddHealth data (kind of restrictive in nature wrt. access, yet nationally representative) to evaluate hypertension risk in young adults. The study is timely, and the Bayesian statistical methods employed seems reasonable, and an intelligent use (in an attempt to model the data hierarchy via random effects, which the Bayesian paradigm elegantly accommodates!). The study was approved by the respetive IRB. My questions are as follows:

(a) AddHealth is a wonderful resource; how are missing data (responses) considered within the proposed CCMMs ?

(b) Any effort in producing a sample size/power statement with a desired effect size wrt the study goals will be highly appreciated, and helpful in determing the size of the analytical data for future analysis. The sample size statement should consider the primary response variable, and possible hierarchy, mimicking AddHealth. If a hierarchical design is not feasible, some ballpark estimates would be helpful.

(c) CCMMs are wonderful tools. However, the CCMM equation appears restrictive in terms of the random error term. Specifically, how can one guarantee that the SBPs are usually Gaussianly distributed? Provide references or arguments is support.

(d) Presence of 2 random effects terms often complicates the analysis. Can authors justify that they are better off putting 2 random effects (instead of one), via a lower DIC value? The data design suggests the hierarchy using random effects; there maybe situations where model fit may suggest otherwise.

Reviewer #2: Abdel Magid and colleagues investigate the association of individual, school and neighboorod influences with blood pressure traits and hypertension in a longitudinal study, the National Longitudinal Study of Adolescent to Adult Health.

Different Cross-classified multilevel models were built to understand whether the health outcomes are influenced by multiple social and physical contexts. I find the methods and data analysis performed in a considerable way.

However some parts of the manuscript should be improve and presented in a clearer way.

1) Abstract: Age range could be included. ICC has not been introduced.

2) "Individual, School, and Neighborhood Variables" the description of which variables are included in the school and neighboorod level should be reported in a clearer way.

3) Page 10, line 268: "were examined using two-sample t-tests for continuous variables and chi-square tests for categorical variables". It would be good to have the p-value added in the Table 1.

4) Page 10 line 280: "were fit using MLwiN (version 3.00; Birmingham, UK) via Stata’s runmlwin command". This part could integrated with the one in line 330.

5) Page 12, line 317: DBP is not written

6) Do the authors think could it be a good idea to represent some results graphically?

7) Pag 13, line 341: Is the number reported as hypertensive correct?

6. PLOS authors have the option to publish the peer review history of their article (what does this mean?). If published, this will include your full peer review and any attached files.

Reviewer #1: No

Reviewer #2: No

---

## [Author Response · Author response to Decision Letter 0]

4 Mar 2022

Responses are in bold and changed text is indicated by italics, below

JOURNAL REQUIREMENTS 

PLOS ONE’s STYLE REQUIRMENTS

• Please ensure that your manuscript meets PLOS ONE's style requirements, including those for file naming. The PLOS ONE style templates can be found at https://journals.plos.org/plosone/s/file?id=wjVg/PLOSOne_formatting_sample_main_body.pdf and https://journals.plos.org/plosone/s/file?id=ba62/PLOSOne_formatting_sample_title_authors_affiliations.pdf

o To adhere to PLOS One’s style requirements, we have made the following changes: bold type 18pt font for level 1 headings, bold type 16pt font for level 2 headings, manuscript title in sentence case, removed author’s titles, spelled out “USA” to “United States of America”, removed corresponding author’s physical address and kept corresponding author’s email. 

CODE SHARING

• Please note that PLOS ONE has specific guidelines on code sharing for submissions in which author-generated code underpins the findings in the manuscript. In these cases, all author-generated code must be made available without restrictions upon publication of the work. Please review our guidelines at https://journals.plos.org/plosone/s/materials-and-software-sharing#loc-sharing-code and ensure that your code is shared in a way that follows best practice and facilitates reproducibility and reuse.

o In accordance with PLOS ONE’s specific guidelines on code sharing we have shared our author-generated code underpinning the findings in this manuscript on Open Science Framework. This code can be found at: https://osf.io/86fpn/?view_only=57084773c8b642a497c1d6c8806ab0ee

COMPETING INTERESTS SECTION

• Thank you for stating the following in the Competing Interests section: "I have read the journal's policy and the authors of this manuscript have the following competing interests: Michelle Odden is a consultant for Cricket Health, Inc. The remaining authors have indicated no conflicts of interest to disclose."We note that you received funding from a commercial source: Cricket Health, Inc Please provide an amended Competing Interests Statement that explicitly states this commercial funder, along with any other relevant declarations relating to employment, consultancy, patents, products in development, marketed products, etc. Within this Competing Interests Statement, please confirm that this does not alter your adherence to all PLOS ONE policies on sharing data and materials by including the following statement: "This does not alter our adherence to PLOS ONE policies on sharing data and materials.” (as detailed online in our guide for authors http://journals.plos.org/plosone/s/competing-interests). If there are restrictions on sharing of data and/or materials, please state these. Please note that we cannot proceed with consideration of your article until this information has been declared. Please include your amended Competing Interests Statement within your cover letter. We will change the online submission form on your behalf

o We confirm that this competing interest does not alter our adherence to all PLOS ONE policies on sharing data and materials. The “Potential Conflicts of Interests” section now reads as: “Potential Conflicts of Interest: Michelle Odden is a consultant for Cricket Health, Inc. The remaining authors have indicated no conflicts of interest to disclose. This does not alter our adherence to PLOS ONE policies on sharing data and materials.” This updated statement is also included in our cover letter.

REVIEWER COMMENTS

REVIEWER #1

• The manuscript considers secondary data analysis and fitting of cross-classified multilevel models (CCMM) to AddHealth data (kind of restrictive in nature wrt. access, yet nationally representative) to evaluate hypertension risk in young adults. The study is timely, and the Bayesian statistical methods employed seems reasonable, and an intelligent use (in an attempt to model the data hierarchy via random effects, which the Bayesian paradigm elegantly accommodates!). The study was approved by the respetive IRB. My questions are as follows:  

o Thank you for your succinct summary and thoughtful, insightful review. We appreciate your careful review of our research.

• (a) AddHealth is a wonderful resource; how are missing data (responses) considered within the proposed CCMMs ?  

o We conducted a complete case analysis and note in the Methods how many indviduals were dropped. Total indviduals in Wave IV is 15,701 and we include 13,911 in our analysis. In the final sample, there was no additional missing data.

• (b) Any effort in producing a sample size/power statement with a desired effect size wrt the study goals will be highly appreciated, and helpful in determing the size of the analytical data for future analysis. The sample size statement should consider the primary response variable, and possible hierarchy, mimicking AddHealth. If a hierarchical design is not feasible, some ballpark estimates would be helpful.

o We found small effect sizes for both school and neighborhood context (intercept variance parameters). A larger sample size may have allowed us to detect small effect sizes, though we were constrained by the Add Health sample size and school and neighborhood structure. Literature on sample size for CCMMs are sparse though one study (Chung 2018; linked below) indicated good credible interval coverage across a wide range of number of higher level units, group sizes within those units, and extent of cross-classification. Per these guidelines, Add Health has a very large number of neighborhoods (n=1900) and a sufficient number of schools (n=128). Future studies using similar datasets with non-hierarchical nesting of students within schools and neighborhoods may have more variability at the contextual level and would therefore require lower sample sizes. Citation: Chung, H., Kim, J., Park, R. and Jean, H., 2018. The impact of sample size in cross-classified multiple membership multilevel models. Journal of Modern Applied Statistical Methods, 17(1), p.26. https://digitalcommons.wayne.edu/cgi/viewcontent.cgi?article=2491&context=jmasm

• (c) CCMMs are wonderful tools. However, the CCMM equation appears restrictive in terms of the random error term. Specifically, how can one guarantee that the SBPs are usually Gaussianly distributed? Provide references or arguments is support. I

o In our sample, SBP and DBP are very normally distributed (mean and median nearly identical, SDs are not large compared to the means, etc.). For example, SBP mean=124.5 (SD=13.6) vs. median=123.5DBP mean=79.0 (SD=10.2) vs. median=78.5. We acknowledge that SBP are not necessarily usually Gausianly distributed in adult populations and usually have a wide range. Moreover, in other samples, it may be necessary to account for non-normality by transforming BP values in order to use linear regression or to use a different distribution in the model. Neverthless, in our specific case the age range is relatively small and constrained to young adulthood where things are a bit less variable/skewed. 

• (d) Presence of 2 random effects terms often complicates the analysis. Can authors justify that they are better off putting 2 random effects (instead of one), via a lower DIC value? The data design suggests the hierarchy using random effects; there maybe situations where model fit may suggest otherwise.

o Though it is usually not advised to compare model fit between multilevel models where the number of levels/model structurure differs (it is okay to compare models with the same structure but differing predictors). Nevertheless, given these models are fit using maximum likelihood, we have included the DIC values in Supplementary Table 3 for the crude cross-classified multilevel models for all outcomes. The best fitting null models were a bit variable, but given the data structure and our research objective of examinining the simultaneous impact of school and neighborhood, we used the cross classified models across all outcomes. Additionally, while the neighborhood ICCs are realtively small, they are not zero, and therefore an important context to account for as we have in our study. Moreover, there is a signfigant amount of literature regarding the importance of comprehensively accountying for the study design such as we have done with the linear and logistic CCMMs. As included in our citation of Dunn et al Health and Place 2015, ignoring one of the levels is equated to misattributing the variance to the level that was included in the analysis (i.e. school-only ‘absorbs’ the neighborhood effect and makes it seem like it is a school level effect. Once you compare to the CCMM model, you can see the variance is parsed between school and neighborhood). The citation below further supports the importance of comprehensively accountying for the study design with CCMMs. While we acknowledge that this matters more and is more apparent when the variance is larger, this is also more apparent in linear models (such as in SBP and DBP models included in this study). Moreover, as shown in supplementary table 3, according to the DIC values the CCMM is best fitting for all outcomes except the 140/90 definition of hypertension. Therefore, this provides further evidence that the CCMMs were better on the whole.

o Citation: Ren, W., 2011. Impact of design features for cross-classified logistic models when the cross-classification structure is ignored. The Ohio State University. https://etd.ohiolink.edu/apexprod/rws_etd/send_file/send?accession=osu1322538958&disposition=inline

REVIEWER #2

• Abdel Magid and colleagues investigate the association of individual, school and neighboorod influences with blood pressure traits and hypertension in a longitudinal study, the National Longitudinal Study of Adolescent to Adult Health. Different Cross-classified multilevel models were built to understand whether the health outcomes are influenced by multiple social and physical contexts. I find the methods and data analysis performed in a considerable way. However some parts of the manuscript should be improve and presented in a clearer way. 

o Thank you for your succinct summary and thoughtful, insightful review. We appreciate your careful review of our research.

• 1) Abstract: Age range could be included. ICC has not been introduced. 

o We have included the following updated sentence in the abstract “Data were analyzed from the National Longitudinal Study of Adolescent to Adult Health (1994-1995 ages 11-18 and 2007-2008 aged 24-32).” The age ranges in each Add Health wave are detailed here: https://addhealth.cpc.unc.edu/documentation/study-design/ We also added the following sentence in the Abstract-Methods section “For linear models, intra-class correlations (ICC) are reported for random effects.

• 2) "Individual, School, and Neighborhood Variables" the description of which variables are included in the school and neighboorod level should be reported in a clearer way.

o We have reported the description of the three levels of variables in three sections in the methods section. This section now reads as: 

“Individual Variables 

We constructed individual covariates using data from the Wave I in-home interview, including adolescents’ biological sex (male, female), race/ethnicity (non-Hispanic Black, Hispanic, Asian and Pacific Islander, Other, Multiracial, and non-Hispanic White). At the individual-level, SES was determined based on parental education and receipt of public assistance. We used data from either the youth or caregiver interview to capture receipt of public assistance (mother currently receiving public assistance, such as welfare or not) and highest level of parental education (defined as the maximum level of education by the resident mother, resident father, or resident step-father/partner (no high school diploma or equivalent; completed high school or equivalent; completed some college, trade school or a 2-year degree; completed equivalent 4-year college degree or above). Height and weight were measured by trained interviewers at Wave IV. Young adult body mass index (BMI) at Wave IV was calculated as the ratio of weight in kilograms over height in meters squared. Age at Wave IV (in years) was calculated from the date of Wave IV in-home interview and participant’s date of birth. 

School Variables 

We constructed school-level covariates using data from the Wave I data. Using the survey of the full sample of schools, at the school-level, we created a continuous measure of school-level SES by aggregating individual-level data. Use of individual-level data was required as information about school-level SES was not directly available. We calculated the proportion of students within each school whose mother had received public assistance or had a college degree. 

Neighborhood Variables 

We constructed neighborhood-level covariates using data from the Wave I data. At the neighborhood level, we used data from the 1990 Census to create a neighborhood-level SES measure indicating the proportion of residents within each neighborhood who had received public assistance or had a college degree. We also calculated the proportion of students in either the school or the neighborhood who were White.”

• 3) Page 10, line 268: "were examined using two-sample t-tests for continuous variables and chi-square tests for categorical variables". It would be good to have the p-value added in the Table 1. 

o We have included p-values in table 1. 

• 4) Page 10 line 280: "were fit using MLwiN (version 3.00; Birmingham, UK) via Stata’s runmlwin command". This part could integrated with the one in line 330. 

o Sentences in 280 and 330 have now been combined at the end of Statistical Analysis section and now reads “Models were fit using MLwiN (version 3.00; Birmingham, UK) via Stata’s runmlwin command. MLwiN uses Bayesian estimation procedures using Markov Chain Monte Carlo (MCMC) methods with non-informative priors and a Metropolis-Hastings sampling algorithm allowing for simultaneous modeling of non-hierarchically nested contexts.24-27All univariate and bivariate analyses were preformed using Stata version 16 (College Station, TX).”

• 5) Page 12, line 317: DBP is not written 

o This sentence now reads: “All models for MAP, DBP, and SBP additionally adjusted for self-reported use of antihypertensive medications.”

• 6) Do the authors think could it be a good idea to represent some results graphically?

o Given that the school and neighborhood variation contributions are less than <1.5%, we do not think representing the results graphically would add much to the results above and beyond what is included in the tables. 

• 7) Pag 13, line 341: Is the number reported as hypertensive correct?

o We have double checked this number, and yes it is correct: “Of the 13,911 Wave IV participants included in this study, 7,111 (51%) young adults were classified as hypertensive.” We would like to note that this number is based on the more consertavitve, newer definition of >130/80. We examined both commonly used thresholds for hypertension included 130/80 and 140/90. The number hypertensive has also been found in the literature, including in this AddHealth study published in PLoS One: Nagata JM, Ganson KT, Cunningham ML, et al. Associations between legal performance-enhancing substance use and future cardiovascular disease risk factors in young adults: A prospective cohort study. PLoS One. 2020;15(12):e0244018. Published 2020 Dec 15. doi:10.1371/journal.pone.0244018.

---

## [Decision Letter · Decision Letter 1]

28 Mar 2022

Adolescent Individual, School, and Neighborhood Influences on Young Adult Hypertension Risk

PONE-D-21-33507R1

Dear Dr. Nagata,

We’re pleased to inform you that your manuscript has been judged scientifically suitable for publication and will be formally accepted for publication once it meets all outstanding technical requirements.

Kind regards,

Giacomo Pucci

Academic Editor

PLOS ONE

**Comments to the Author**

1. If the authors have adequately addressed your comments raised in a previous round of review and you feel that this manuscript is now acceptable for publication, you may indicate that here to bypass the “Comments to the Author” section, enter your conflict of interest statement in the “Confidential to Editor” section, and submit your "Accept" recommendation.

Reviewer #1: All comments have been addressed

Reviewer #2: All comments have been addressed

2. Is the manuscript technically sound, and do the data support the conclusions?

Reviewer #1: (No Response)

Reviewer #2: Yes

3. Has the statistical analysis been performed appropriately and rigorously? 

Reviewer #1: (No Response)

Reviewer #2: Yes

4. Have the authors made all data underlying the findings in their manuscript fully available?

Reviewer #1: (No Response)

Reviewer #2: Yes

5. Is the manuscript presented in an intelligible fashion and written in standard English?

Reviewer #1: (No Response)

Reviewer #2: Yes

6. Review Comments to the Author

Reviewer #1: (No Response)

Reviewer #2: Thanks to the authors to answer clearly and exhaustively all my comments.

I have no further issue to raise.

7. PLOS authors have the option to publish the peer review history of their article (what does this mean?). If published, this will include your full peer review and any attached files.

Reviewer #1: No

Reviewer #2: No

---

## [Editor Report · Acceptance letter]

5 Apr 2022

PONE-D-21-33507R1 

Adolescent individual, school, and neighborhood influences on young adult hypertension risk 

Dear Dr. Nagata:

I'm pleased to inform you that your manuscript has been deemed suitable for publication in PLOS ONE. Congratulations! Your manuscript is now with our production department. 

Kind regards, 

on behalf of

Dr. Giacomo Pucci 

Academic Editor

PLOS ONE